# Cultured Human Foreskin as a Model System for Evaluating Ionizing Radiation-Induced Skin Injury

**DOI:** 10.3390/ijms23179830

**Published:** 2022-08-29

**Authors:** Yanick Hippchen, Gargi Tewary, Daniela Jung, Zoé Schmal, Stephan Meessen, Jan Palm, Claudia E. Rübe

**Affiliations:** 1Department of Radiation Oncology, Saarland University Medical Center, 66421 Homburg, Germany; 2Department of Urology, Klinikum Saarbrücken, 66119 Saarbrücken, Germany

**Keywords:** cultured foreskin, ionizing radiation, radiation-induced dermatitis, histone variant H2A.J, cellular senescence, senescence-associated secretory phenotype (SASP)

## Abstract

Purpose: Precise molecular and cellular mechanisms of radiation-induced dermatitis are incompletely understood. Histone variant H2A.J is associated with cellular senescence and modulates senescence-associated secretory phenotype (SASP) after DNA-damaging insults, such as ionizing radiation (IR). Using ex vivo irradiated cultured foreskin, H2A.J was analyzed as a biomarker of radiation-induced senescence, potentially initiating the inflammatory cascade of radiation-induced skin injury. Methods: Human foreskin explants were collected from young donors, irradiated ex vivo with 10 Gy, and cultured in air-liquid interphase for up to 72 h. At different time-points after ex vivo IR exposure, the foreskin epidermis was analyzed for proliferation and senescence markers by immunofluorescence and immunohistochemical staining of sectioned tissue. Secretion of cytokines was measured in supernatants by ELISA. Using our mouse model with fractionated in vivo irradiation, H2A.J expression was analyzed in epidermal stem/progenitor cell populations localized in different regions of murine hair follicles (HF). Results: Non-vascularized foreskin explants preserved their tissue homeostasis up to 72 h (even after IR exposure), but already non-irradiated foreskin epithelium expressed high levels of H2A.J in all epidermal layers and secreted high amounts of cytokines. Unexpectedly, no further increase in H2A.J expression and no obvious upregulation of cytokine secretion was observed in the foreskin epidermis after ex vivo IR. Undifferentiated keratinocytes in murine HF regions, by contrast, revealed low H2A.J expression in non-irradiated skin and significant radiation-induced H2A.J upregulations at different time-points after IR exposure. Based on its staining characteristics, we presume that H2A.J may have previously underestimated the importance of the epigenetic regulation of keratinocyte maturation. Conclusions: Cultured foreskin characterized by highly keratinized epithelium and specific immunological features is not an appropriate model for studying H2A.J-associated tissue reactions during radiation-induced dermatitis.

## 1. Introduction

Radiotherapy is an important oncological treatment modality whereby ionizing radiation (IR) is locally delivered to eradicate cancer. Even with the technological advancement in radiotherapy planning and application, high percentages of patients receiving fractionated radiotherapy still suffer from radiation-induced skin toxicity, ranging from mild erythema and dry desquamation to more severe moist desquamation and up to skin ulceration [1]. However, the precise mechanism of radiation-induced skin injury at therapeutically relevant doses remains incompletely understood.

Healthy skin provides a protective barrier between the body and its environment for one’s entire life [2]. The outermost epidermis is stratified into four layers, with keratinocytes being the predominant cell type. The *stratum basale* represents a single-cell layer attached to the basal lamina and comprises epidermal stem/progenitor cells. Basal cells and deep cells within the *stratum spinosum* undergo mitosis, thereby forming new keratinocytes that migrate into more superficial layers and complete their physiological maturation. By the time keratinocytes reach the *stratum granulosum*, they flatten and produce more keratin filaments and lipid-filled membrane-coated vesicles, thereby forming water-resistant barriers. The *stratum corneum* consists of numerous layers of flattened, dead corneocytes lacking nuclei. During their constant differentiation, keratinocytes migrate from basal to cornified layers and finally desquamate from the skin surface. Epidermal homeostasis depends on the regenerative capacity of stem/progenitor cells and on epigenetic mechanisms that regulate the differentiation and maturation processes in keratinocytes. The dermis comprises the superficial papillary layer and the deep, thicker reticular layer. The primary cell type in the dermis are fibroblasts, which produce extracellular structural proteins, such as collagen and elastin. The dermis contains hair follicles, sweat, sebaceous and apocrine glands, and blood vessels, providing nourishment and waste removal for both dermal and epidermal cells [3]. Even if the basic histological structure is identical, the composition of human skin is generally adapted to the particular requirements of the anatomical regions [4]. Foreskin consists of a double-sided layer of stratified squamous epithelium, with thickened and more keratinized epidermal layer and increased immune cell densities compared to other skin tissues, providing a robust physical barrier to mechanical and other environmental stress [5].

Since neither 3D-tissue organization nor multiple cell-type composition is reproduced in monolayer single cell-type cultures, our laboratory pioneered the use of foreskin biopsies to study the skin tissue response after ex vivo IR exposure [6]. Cellular stress responses induced by IR are highly complex. Increasing evidence suggests that senescent cells accumulate in the skin after IR exposure and may contribute to radiation-induced skin pathologies [7]. Senescent cells are characterized by their inability to proliferate, their resistance to apoptosis, and the secretion of factors that promote inflammation and tissue deterioration [8,9]. Previous studies have shown that histone variant H2A.J accumulates in human fibroblasts during replicative and radiation-induced senescence and modulates the secretion of inflammatory cytokines [10,11]. Our previous research shows that H2A.J is a sensitive marker of senescent keratinocytes in the human epidermis in the context of skin aging [12]. Here, we aimed to analyze radiation-induced senescence in human skin by establishing cultured foreskin explants for ex vivo IR exposure to mimic the complex radiation responses.

## 2. Results

### 2.1. H2A.J and Ki67 Expression in Foreskin Explants after Ex Vivo Irradiation

To test histone variant H2A.J as a biomarker for radiation-induced senescence, cultured foreskin samples were stained for H2A.J to analyze keratinocyte senescence (Figure 1). Already in the non-irradiated foreskin, the majority of keratinocytes were highly positive for H2A.J by IHC staining; after ex vivo irradiation, there was a tendency for stronger H2A.J staining intensity (Figure 1A). However, immunofluorescence microscopy of non-irradiated foreskin explants showed different H2A.J expression levels in the various epidermis layers (Figure 1B). Accordingly, relative proportions of H2A.J+ keratinocytes were analyzed in the different epidermal layers (Figure 1A). Independently of IR exposure, we observed the highest proportions of H2A.J+ keratinocytes (≈80%) in granular layers, where epidermal keratinocytes are transformed from living cells into corneocytes (Figure 1C). Compared to the granular layer, we observed clearly lower H2A.J expression levels in the *stratum spinosum* and *basale* (≈50%), reflecting epidermal layers with less differentiated keratinocytes. To explore the role of H2A.J for induction of cellular senescence, cultured foreskin samples were exposed to IR (10Gy) and analyzed 24h after ex vivo exposure. Unexpectedly, our results revealed no radiation-dependent increase in H2A.J+ cells in the different epidermal layers (Figure 1C). Accordingly, our findings suggest that tissue-specific H2A.J expression in the foreskin is independent of senescence induction after IR exposure. Using the proliferation marker Ki67, we examined the regenerative capacity in human foreskin samples in relation to IR exposure (Figure 1B). In non-irradiated foreskin, Ki67+ cells were most abundant in the *stratum spinosum* (6% ± 0.1%), less frequent in the *stratum basale* (2% ± 0.1%) and completely absent in the *stratum granulosum* (Figure 1C). After IR exposure with 10Gy, the Ki67+ cell fractions in the different epidermal layers remained stable, suggesting that the proliferative capacity of the foreskin epidermis is not affected by these IR doses. Moreover, using Ki67 in direct combination with H2A.J, we constantly observed an inverse correlation between Ki67+ and H2A.J+ cells (Figure 1B). The fact that Ki67-H2A.J immunostaining was mutually exclusive strongly suggests that cells expressing H2A.J have lost their ability to proliferate.

### 2.2. Individual H2A.J Expression in Foreskin Explants after Ex Vivo Irradiation

The possibility of identifying patient-related factors that are associated with individual radiosensitivity would optimize adjuvant treatment by radiotherapy, limiting the severity of normal tissue reactions. In this study, we aimed to test whether H2A.J-associated inflammation could serve as a possible biomarker of individual radiosensitivity. Accordingly, a larger number of foreskin specimens was screened without any differences in H2A.J expression being observed. Only specimens from young donors were quantitatively evaluated, as we assumed that foreskin samples from children would have very low baseline H2A.J expression due to their young age. In Figure 2, the H2A.J+ and Ki67+ cells before and following IR exposure are shown for each individual donor in order to detect possible inter-individual differences. However, our cohort of young donors showed no differences in terms of either baseline or radiation-induced H2A.J expression (Figure 2A). Moreover, p21 staining of non-irradiated and irradiated foreskin explants revealed only very low levels of p21-positive cells, suggesting that at this dose and examination time, there is no induction of premature senescence in foreskin keratinocytes (Appendix A).

### 2.3. Individual Cytokine Secretion in Foreskin Explants after Ex Vivo Irradiation

Recent progress in molecular radiobiology has improved the mechanistic understanding of normal-tissue effects and shifted the focus from initial-damage induction to the concerted biological response at cell and tissue levels affected by early activation of cytokine cascades. Radiation-induced skin inflammation is characterized by an almost immediate upregulation of pro-inflammatory cytokines, thereby recruiting inflammatory cells from the surrounding tissue. Here, we measured cytokine expression levels in foreskin supernatants for interleukins 6 and 8 (IL-1 and IL-6) and monocyte chemoattractant protein-1 (MCP-1), representing H2A.J-associated signature cytokines [10,11]. Our findings suggest that foreskin samples from young donors vary in their inflammatory response to IR exposure (Figure 3). For the donor KNN, we observed significantly elevated cytokine expression levels for IL-6, IL-8, and MCP-1 following IR exposure (Figure 3); this finding may suggest that this individual develops stronger inflammatory reactions in response to radiotherapy.

### 2.4. H2A.J Expression in Murine Skin after Fractionated In Vivo Irradiation

The long-term function of non-vascularized skin explants during in vitro culturing is particularly limited by the maintenance of epidermal stem cell compartments. Human skin regions from different anatomical locations are characterized by the same basic histological structure but vary considerably in the thickness of their layers and their specific adnexal structures. After injury, skin appendages such as hair follicles (containing hair follicle stem cells in bulge regions) are capable of re-epithelialization via the migration of keratinocytes from the adnexal epithelium to the surface of the epidermis [13]. Upon skin injury, multiple stem cell populations in hair follicles and interfollicular epidermis make distinct contributions to the regenerating epithelium over time [13,14]. To overcome tissue-culture-associated limitations, we used our mouse model of fractionated in vivo irradiation to investigate H2A.J and cytokine expression in murine skin over a longer period, ultimately to recapitulate radiation-induced skin toxicities more closely. Accordingly, laboratory mice were exposed to fractionated irradiation (5 × 2 Gy) and at defined time-points (72 h, 1, and 2 weeks) after the last in vivo exposure, the back-skin epidermis was harvested for analysis of H2A.J expression and cytokine secretion. Figure 4A shows representative pictures of H2A.J- and SA-ß-Gal expression in hair follicles in non-irradiated versus irradiated dorsal skin (1w post-IR). Subsequently, H2A.J+ and Sa-ß-Gal+ cells were quantified in different regions (infundibulum, bulge, papilla) of hair follicles in murine skin sections. While non-irradiated skin revealed very low numbers of H2A.J+ cells (≈2%), the percentage of H2A.J+ cells increased significantly after fractionated in vivo irradiation, from ≈40% at 72 h post-IR to ≈70% at 2 w post-IR. The number of SA-ß-Gal+ cells also increased significantly after IR exposure, but to clearly lower levels, from ≈1% in non-irradiated skin to ≈16% at 72 h post-IR and ≈40% at 2 w post-IR. These findings suggest that H2A.J expression in different regions of hair follicles cannot be equated with senescent keratinocytes. However, we observed slight differences in radiation-induced H2A.J expression between the different hair follicle regions, with the lowest expression in bulge regions as putative hair follicle stem cells niche. Hair follicles stem cells are characterized by their long-term potential for hair follicle regeneration. Previous studies have unraveled the specific role of histone modifications in maintaining the epigenetic control between stem cell quiescence and differentiation in skin homeostasis [15,16]. Based on our findings, we presume that differential H2A.J expression in different epidermal cell populations and the increased H2A.J expression after IR exposure may reflect an epigenetic regulation mechanism inducing an epidermal differentiation process in keratinocytes.

To evaluate the inflammatory effects of fractionated in vivo irradiation, we analyzed the secretion of different cytokines in skin supernatants by ELISA. Cytokines analysis of IL-6 and MCP-1 revealed slightly higher expression levels at 72 h and 1 w post-IR compared to non-irradiated controls, suggesting transient inflammatory effects after this fractionated in vivo irradiation, inducing no major damage in epidermal keratinocytes. Interleukin-8 has important functions in initiating inflammation in humans, attracting immune cells such as neutrophils, thereby contributing to chronic inflammation. However, mice normally do not have the IL-8 gene [17]; thus, our IL-8 measurements may reflect non-specific detection of other components.

## 3. Discussion

The foreskin serves as the primary barrier to many pathogens and therefore has a very characteristic epidermal structure with specific immunological properties [18]. The thickened and more cornified foreskin epithelium is associated with highly increased H2A.J expression (even without IR exposure), potentially reflecting stronger differentiation and maturation processes to build up trans-epithelial resistance. The natural state of the male foreskin is characterized by chronic inflammatory milieu; this is in line with our observation of clearly increased secretion of inflammatory cytokines (already without IR exposure) [5]. In largely undifferentiated epidermal cell populations of murine hair follicles, by contrast, the basic expression of H2A.J was very low and only enhanced with increasing time after IR exposure. Direct comparison with SA-ß-Gal staining in different skin areas suggests that H2A.J is not a specific marker for cellular senescence but rather a marker for epithelial differentiation [19]. Based on our findings, we presume that differential H2A.J expression in various epidermal cell populations and increased H2A.J expression after IR exposure may reflect epigenetic regulation mechanisms inducing differentiation and maturation processes in keratinocytes.

Over recent years, 3D co-culturing methodologies of skin equivalents have been developed to mimic human skin pathology more accurately than commonly used 2D monolayer cell cultures [6]. Moreover, the possibility of generating cultured explants from patient-derived skin opened a new perspective to investigate individual radiation-induced skin reactions, thereby improving efficient translational research and personalized treatments [20]. Skin explants can be feasible solutions to address the missing link between the conditions in vivo and oversimplified 2D monolayer or 3D co-culturing models (such as spheroids or organoids) in vitro. Here, we used human foreskin explants to test H2A.J as a biological marker to measure senescent keratinocytes after IR exposure.

Predictive assays testing individual radiosensitivity would greatly help to establish personalized risk specifications for radiotherapy, with the perspective of modifying the treatment in radiosensitive individuals. However, our findings suggest that H2A.J expression is not a feasible marker for radiation-induced senescence in foreskin explants and cannot predict radiation-induced skin toxicity.

Previous in vitro studies indicate that H2A.J incorporation into the chromatin of senescent cells leads to epigenetic modifications and is associated with the secretion of inflammatory cytokines, collectively known as SASP [10,11]. In previous studies investigating abdominal skin biopsies from adult donors before and after ex vivo IR exposure, we could show that the H2A.J expression was significantly increased in keratinocytes of irradiated skin, suggesting that H2A.J may be a promising marker for radiation-induced senescence and may predict radiotherapy responses [12]. Here, analyzing foreskin explants from young donors (1–9 years), we observed extremely high H2A.J expression levels in all epidermal layers, even without genotoxic stress by IR exposure. This observation is in sharp contrast to our previous results, showing only low H2A.J expression levels (≈20%) in the abdominal epidermis of young donors (≤20 years) but highly increased H2A.J expression levels (≈70%) in aged donors (≥60 years) [12]. Moreover, analyzing foreskin explants, we found no radiation-induced increase in H2A.J or cytokine expression levels. Collectively, H2A.J expression and cytokine secretion in keratinocytes of human foreskin explants were dramatically higher -before and after IR exposure- than in other human body skin or in undifferentiated keratinocytes of murine hair follicles. Based on these findings, we presume that histone variant H2A.J is associated with the epigenetic regulation of epidermal keratinocytes during their progression from the undifferentiated basal to the differentiated state. Moreover, increased H2A.J expression in relatively undifferentiated keratinocytes after IR exposure may reflect the early activation of differentiation processes in response to DNA damage. Genotoxic stress triggers cascades of inflammatory signaling pathways, leading to the release of pro-inflammatory factors and thereby modulating immune responses [21]. Further mechanistic understanding of DNA damage-induced immunomodulatory responses on skin homeostasis after IR exposure might shed light on the complex pathophysiology of radiation-induced dermatitis.

Dynamic epigenetic changes allow keratinocytes to regulate their gene expression programs to fulfill specialized cellular functions within different epidermal layers. Upon their maturation process, differentiated keratinocytes exit the cell cycle and migrate from the basal layer through the more superficial spinous and granular layers, culminating in their cornification and cell death. How keratinocytes dynamically govern the hierarchy of self-renewal, differentiation, and maturation to finely balance proliferation and differentiation processes is still poorly understood [13,22,23]. Keratinocyte function depends on complex gene regulatory networks that ultimately determine gene expression changes and guide the transition from basal to differentiated keratinocyte states. Epigenetic modifications such as the incorporation of histone variant H2A.J may impact the transcriptional regulation of keratinocytes and are likely coupled to the spatial organization of the epidermis. We observed substantial differences between the abdominal and foreskin epidermis in terms of thickness and level of keratinization, likely reflecting their skin-specific exposures to mechanical stress and environmental stimuli [5]. Based on the characteristic staining patterns in different epidermal keratinocytes, we presume that H2A.J expression may have a previously unappreciated role in epigenetic regulation of keratinocyte transition from basal to differentiated states.

In radiotherapy, the threshold range for transient erythema of the skin is 3–5 Gy, and prolonged erythema has threshold ranges of 5–10 Gy. We intended the application of a dose that evokes an inflammatory response but does not damage the foreskin explant too much by radiation-induced apoptosis or necrosis. In the mouse model, fractionated irradiation was carried out in vivo in accordance with the clinical conditions of patients in radiotherapy. This prolonged irradiation schedule is only possible in vascularized mouse skin in vivo but not in non-vascularized foreskin explants in vitro since its cultivation period is limited to a few days. The biological effect of a single dose of 10 Gy and fractionated irradiation of 5 × 2 Gy (=10 Gy) is indeed very different. Despite this lower biological effect, we observed the radiation-induced upregulation of H2A.J in the largely undifferentiated keratinocytes of the hair follicles of mouse skin during fractionated irradiation.

In addition to providing a physical barrier to the outside world, the skin is also an active immune organ, comprised of highly complex networks of innate and adaptive immune cells, whereby different anatomical skin areas have specific immunologic features [13]. The foreskin may develop microtrauma during sexual activity, with disrupted epithelial barriers permitting the entry of pathogens. Previous studies on foreskin pathophysiology described clear differences to normal body skin regarding the cornified envelope, structures of tight junctions, densities of infiltrating immune cells, and secretion of inflammatory cytokines [5]. Together these different features regulate the barrier function of the pluri-stratified foreskin and may explain its increased epithelial resistance upon IR exposure.

Communication between damaged or senescent keratinocytes and their microenvironment plays an important role in the pathophysiology of skin inflammation [21]. However, complex tissue organization with the interaction between the epidermal and dermal compartment and their specific immune responses cannot faithfully be reproduced using foreskin explants. Skin reconstructs mimicking epidermis and dermis similar to in vivo conditions have to be developed to study the pathophysiology of radiation-induced skin injury. However, challenges remain in the manufacturing of skin reconstructs that recreate physiologically relevant microenvironments and encompass main skin cells, incorporate perfused vasculature and skin appendages, as well as components of the immune system.

## 4. Materials and Methods

### 4.1. Human Foreskin Collection

Foreskin samples were collected from young, healthy volunteers (1–9 years, *n* = 12) with medical indications for circumcision. Protocol procedures were approved by local ethics committee (“Ethikkommission der Ärztekammer des Saarlandes”). The parents of study participants provided written informed consent for the collection and analysis of normally discarded foreskin tissues.

### 4.2. Ex Vivo Irradiation of Cultured Foreskin Explants

Foreskin explants were rinsed with PBS and divided into small pieces (0.5 cm^2^). Foreskin samples were incubated dermal side down on polyethylene membranes and epidermal side exposed to air in 6-well plates (each well filled with 2 mL medium) at 37 °C under 5% CO_2_. The culture medium consisted of DMEM, 10,000 units of penicillin, 10 mg/mL streptomycin, and 200 mM/L glutamine. After air-medium interface cultivation for 24 h, foreskin explants were exposed to IR with 10 Gy (6-MV photons, 2 Gy/min) using the linear accelerator Artiste™ (Siemens, Munich, Germany). Twenty-four hours after IR exposure, foreskin tissue samples were embedded for microscopic analysis, and the supernatant was used for cytokine measurement (Appendix A). Histopathological studies confirmed the viability and functionality of our foreskin culture method (Appendix A). Moreover, for different culture periods, we observed no variations in H2A.J+ cell numbers, suggesting that foreskin keratinocytes maintain their epigenetic status (Appendix A).

### 4.3. Immunohistochemistry (IHC)

Formalin-fixed foreskin samples were embedded in paraffin and sectioned at 4 µm thickness. After dewaxing in xylene and rehydration in decreasing concentrations of alcohol, antigen retrieval was performed in citrate buffer, and sections were incubated with anti-H2A.J-antibody (Active Motif, 61793, Carlsbad, CA, USA) followed by biotin-labeled antibodies (Dako, Glostrup, Denmark). Staining was completed by incubation with 3,3′-diaminobenzidine and substrate chromogen. Finally, sections were counterstained with hematoxylin and mounted with Aqueous Mounting Medium (Dako, Glostrup, Denmark). For the visualization of connective tissue, the Masson-Goldner trichromic staining technique was carried out according to the manufacturer’s instructions (Masson-Goldner staining kit, Merck, Darmstadt, Germany).

### 4.4. Immunofluorescence Microscopy (IFM)

Formalin-fixed foreskin tissues were embedded in paraffin and sectioned to 4 µm thickness. After dewaxing in xylene and rehydration in decreasing concentrations of alcohol, sections were boiled in citrate buffer and blocked with 2% goat serum (Carl Roth, Karlsruhe, Germany). Sections were incubated with primary antibodies (anti-H2A.J, Active Motive; anti-Ki67, Abcam, Berlin, Germany), followed by AlexaFluor-488 or AlexaFluor-568 secondary antibodies (Invitrogen Waltham, MA, USA). Finally, sections were mounted in VECTAshield™ with 4′,6-diamidino-2-phenylindole (DAPI; Vector Laboratories, Burlingame, CA, USA). For quantitative analysis ≥1000 epidermal cells were registered for each sample, and H2A.J+ and Ki67+ cells were counted using Nikon E600 epifluorescent microscope (Nikon, Düsseldorf, Germany). To analyze the distribution of H2A.J in different epidermal layers, H2A.J+ cells were enumerated separately in *stratum granulosum spinosum/basale*, and the relative amounts of H2A.J+ cells were depicted per layer.

### 4.5. Fractionated In Vivo Irradiation of Mice

Eight-week-old male C57BL/6 mice (Charles River Laboratories, Sulzfeld, Germany) were housed in groups in IVC cages under standard laboratory conditions. Whole-body irradiation with 5 fractions of 2 Gy (daily IR exposure from Monday to Friday) was performed at the linear accelerator (Artiste™, Siemens), as described previously [24]. Then, 72 h, 1, and 2 weeks after the last IR exposure (sham-)irradiated animals (*n* = 3 per time-point and treatment-group, 18 animals in total) were anesthetized intraperitoneally using Ketamine and Rompun prior to tissue collection. Experimental studies were approved by the Medical Sciences Animal Care and Use Committee of Saarland.

### 4.6. SA-β-Gal Staining

Skin samples were dissected from the back of mice, fixed, and stained overnight with X-Gal solution according to the manufacturer’s instructions (Merck Millipore, MA, USA). After overnight incubation, skin samples were washed, fixed in 4% PFA, and embedded in paraffin. Embedded skin tissue was further processed as already described above for IHC. Senescent cells were identified as blue-stained cells under light microscopy.

### 4.7. Enzyme-Linked ImmunoSorbent Analysis (ELISA)

Supernatants (2 mL per well) were collected from cultured (non-irradiated and irradiated) human foreskin or murine skin samples (sized 0.5 cm^2^) and were frozen at −80 °C until analysis. ELISA kits available for purchase (human: Invitrogen, KHC-0061, KHC-0081, BMS281, Carlsbad, CA, USA; murine: Abcam, ab100713, ab208979, Cambridge, UK; MyBioSource, MBS7606860, San Diego, CA, USA) were used to screen the SASP factors IL-6, IL-8, and MCP-1 with standard ELISA plate reader, according to the supplier’s protocol.

### 4.8. Statistical Analysis

Statistical analyses were performed by unpaired t-test using Graphpad Prism software (Version 8, GraphPad Software, San Diego, CA, USA). Statistical significance was presented as * *p* < 0.05, ** *p* < 0.01, and *** *p* < 0.001.

## Figures and Tables

**Figure 1 ijms-23-09830-f001:**
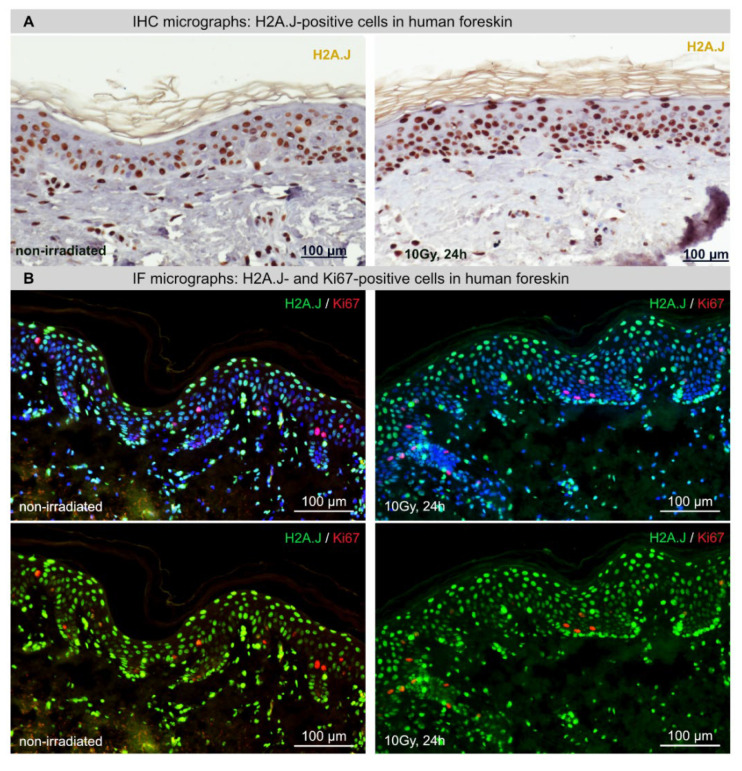
H2A.J and Ki67 expression in foreskin explants after ex vivo IR exposure. (**A**) IHC micrographs of H2A.J staining in foreskin explants after IR exposure (10 Gy; 24 h post-IR) compared to non-irradiated control. (**B**) IF micrographs of H2A.J and Ki67 double-staining in foreskin explants after IR exposure (10 Gy; 24 h post-IR) compared to non-irradiated control. Top photos contain DAPI signals, and bottom photos show the same photos but without corresponding DAPI signals so that the different H2A.J staining intensities (green signal) in the epidermis become more visible. (**C**) Graphic presentation of the quantification of H2A.J+ and Ki67+ cells in epidermis of foreskin explants. Data are presented as mean ± SD (*n* = 12).

**Figure 2 ijms-23-09830-f002:**
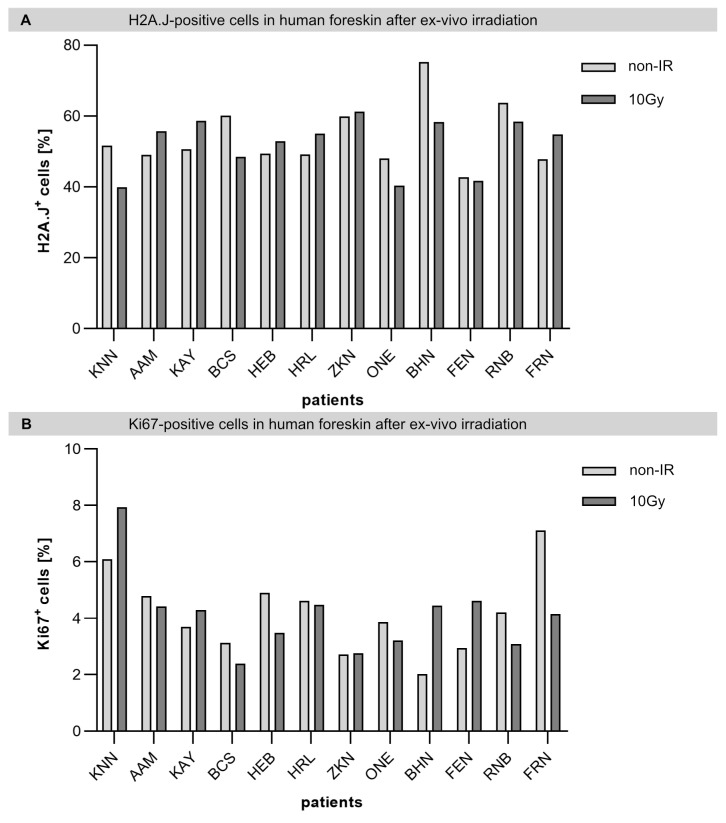
Patient-to-patient variability of H2A.J and Ki67 expression in foreskin explants before and after ex vivo IR exposure. Graphic presentation of the quantification of H2A.J+ (**A**) and Ki67+ keratinocytes (**B**) in the epidermis of foreskin explants before and after IR exposure (10Gy; 24 h post-IR) for the individual donors (listed with the initials of their names) (*n* = 12).

**Figure 3 ijms-23-09830-f003:**
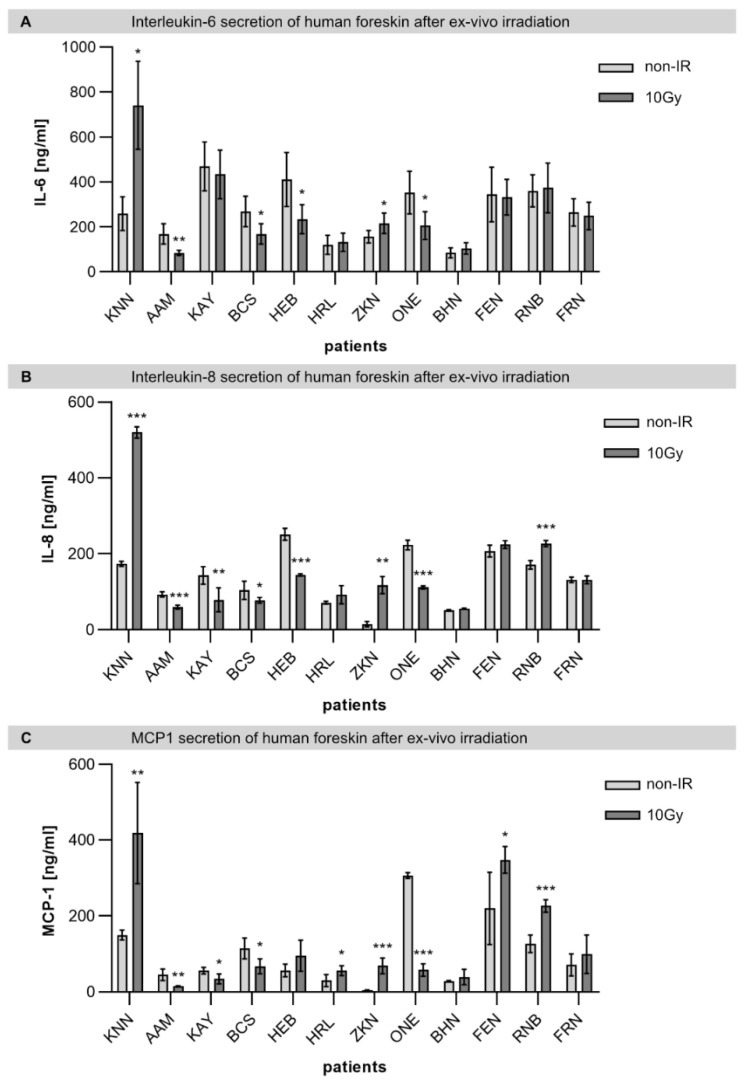
Patient-to-patient variability of cytokine expression in foreskin explants before and after ex vivo IR exposure. Graphic presentation of the quantification of IL-6 (**A**), IL-8 (**B**), and MCP-1 (**C**) measured by ELISA in the supernatant of foreskin explants before (non-IR) and after IR exposure (10 Gy; 24 h post-IR) for individual donors (*n* = 12). * *p* < 0.1; ** *p* < 0.01; *** *p* < 0.001.

**Figure 4 ijms-23-09830-f004:**
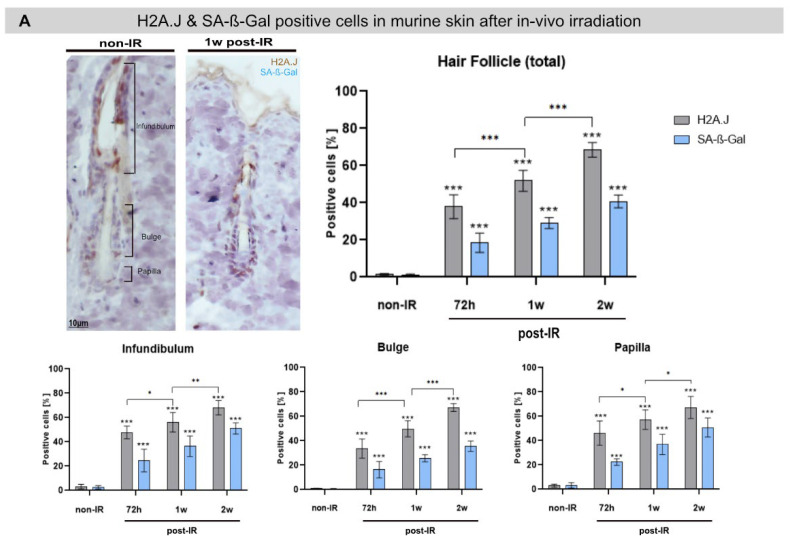
H2A.J and SA-ß-Gal expression in murine skin before and after fractionated in vivo IR exposure. (**A**) Graphic presentation of the quantification of H2A.J+ and SA-ß-Gal+ cells in different regions of murine hair follicles (infundibulum, bulge, papilla) before (non-IR) and at different time-points after fractionated in vivo irradiation (72 h, 1 w, and 2 w post-IR). * *p* < 0.1; ** *p* < 0.01; *** *p* < 0.001 (**B**) Graphic presentation of the quantification of IL-6, IL-8, and MCP-1 measured by ELISA in the supernatant of murine skin before (non-IR) and after fractionated in vivo irradiation (5 × 2Gy; 72 h, 1 w, and 2 w post-IR). Data are presented as mean ± SD (*n* = 3).

## Data Availability

The data that support the findings of this study are available from Saarland University Hospital, Homburg/Saar, Germany.

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
