# Peer review of "Cultured Human Foreskin as a Model System for Evaluating Ionizing Radiation-Induced Skin Injury"

_ijms, 2022, doi:10.3390/ijms23179830_

Round 1

Reviewer 1 Report

The authors are examining the validity of using cultured human foreskin as a model system for evaluating ionizing radiation-induced skin injury.

 However, it is questionable whether the validity can really be evaluated under this experimental condition.

1. Please clarify why you chose a radiation dose of 10 Gy for foreskin.

2. Mice are given 2 Gy in 5 fractions. Please clarify why you gave 2 Gy in 5 fractions to mice.

3. The authors compare results in Foreskin and mice. 2Gy, 5 times and 10Gy single doses have the same total dose of 10Gy, but the biological significance is clearly different. Please explain why you chose this condition, and why you are making comparisons in the discussion.

4. Fig1 is stained with Ki67. Ki67 is commonly used as a marker to determine the malignancy of cancer. Please clarify what you mean by using this as a marker.

5. Please add an explanation for the upper and lower photos of B in Fig1.

6. Please add an explanation about the horizontal axis of Fig2 and Fig3.

Author Response

Reviewer #1

The authors are examining the validity of using cultured human foreskin as a model system for evaluating ionizing radiation-induced skin injury.

However, it is questionable whether the validity can really be evaluated under this experimental condition.

  1. Please clarify why you chose a radiation dose of 10 Gy for foreskin.

In radiotherapy the threshold range for transient erythema is 3-5 Gy, prolonged have threshold ranges of 5-10 Gy. We intended the application of a dose that evokes an inflammatory response, but does not damage the foreskin explant too much by radiation-induced apoptosis or necrosis.

  1. Mice are given 2 Gy in 5 fractions. Please clarify why you gave 2 Gy in 5 fractions to mice.

In the mouse model, fractionated irradiation was be carried out in-vivo in accordance with the clinical conditions for patients in radiotherapy. This prolonged irradiation is only possible in vascularized mouse skin in-vivo, but not in de-vascularized foreskin explants, since its cultivation period is limited to a few days.

  1. The authors compare results in Foreskin and mice. 2Gy, 5 times and 10Gy single doses have the same total dose of 10Gy, but the biological significance is clearly different. Please explain why you chose this condition, and why you are making comparisons in the discussion.

The biological effect of a single dose of 10 Gy and fractionated irradiation of 5x 2Gy is actually very different. Despite this lower biological effect, we observed the radiation-induced upregulation of H2A.J in the largely undifferentiated keratinocytes of the hair follicles during fractionated irradiation.

  1. Fig1 is stained with Ki67. Ki67 is commonly used as a marker to determine the malignancy of cancer. Please clarify what you mean by using this as a marker.

Ki67 is an established proliferation marker. The irradiation itself, but also the cultivation time of the skin explants can influence the proliferation rate of the epidermis. Therefore, we used Ki67 as an internal control for foreskin explant viability.

  1. Please add an explanation for the upper and lower photos of B in Fig1.

In Fig. 1B, the top photos contain the DAPI signal, and the bottom photos show the same photos but without the corresponding DAPI signal. By switching off the DAPI channel, the different H2A.J staining intensities (green signal) in the epidermis becomes more visible.

  1. Please add an explanation about the horizontal axis of Fig2 and Fig3.

On the horizontal axis of Fig. 2 and Fig. 3, the individual patients are listed with the initials of their names.

Reviewer 2 Report

Interesting and relevant article in the area of preclinical study of сultured human foreskin as a model system for evaluating ionizing radiation-induced skin injury. The authors showed that cultured foreskin characterized by highly keratinized epithelium and specific immunological features is not an appropriate model for studying H2A.J-associated tissue reactions during radiation-induced dermatitis. I don't have any questions or comments. The article can be published.

The selected topic of the article is original and relevant in the study. Determination of the model the skin damage caused by ionizing radiation will help in understanding the pathogenesis and search for treatment of radiation injuries.

Studies to determine the model the skin damage caused by ionizing radiation are single. The presented model is described in the article for the first time.

Research methods performed well. Additional research is not required.

In the article, the conclusions with evidence and arguments logically follow from the presented study and correspond to the purpose.

References correspond to the topic and are logically justified

The figures are presented informatively and correspond to the description in the article.

Author Response

Thank you for the positive assessment of our work.

Reviewer 3 Report

This is an intriguing and well-designed study to validate the cultured human foreskin as a model system for evaluating IR-induced skin injury. Basically, this is a follow-up study of the authors who previously characterized a histone variant H2A.J as a senescence marker. In this study, they analyzed whether cultured human foreskin is compatible with the model system for evaluating IR-induced skin aging or injury. Although the conclusion of the study is negative, most results are supported well by experimental data except for Figure 1.

In Figure1, to test H2AJ as a biomarker for IR-induced senescence they co-stained foreskin samples with H2AJ and Ki67. Because reduction in Ki67 expression not necessarily means the cells are undergoing senescence, the authors should use more appropriate senescence markers such as p16 and p21. These two proteins are well known as hallmarks of cellular senescence. Alternatively, the SA-b-gal staining profile upon IR can be presented to validate the IR used in this study can indeed induce cellular senescence. 

Author Response

We used double-staining for H2A.J and Ki67 to show that proliferating Ki67-positive cells are negative for H2A.J and, vice versa, that H2A.J-positive cells are negative for Ki67. These findings support our conclusion that H2A.J labels more differentiated keratinocytes that are no longer dividing. However, our experimental studies with relatively low doses of IR revealed that H2A.J is not a specific marker for cellular senescence in human and murine epidermis; accordingly we observed no clear co-localization between H2A.J and established senescence markers such as p16 or p21.

Round 2

Reviewer 3 Report

Although the authors have had a negative result of p16 and p21 expression as they mentioned in the rebuttal letter, these results are valuable to be presented at least as a supplemental data. 

Author Response

In Suppl. 3 we provide now micrographs of the p21 expression (IFM and IHC) in non-irradiated and irradiated human foreskin. 
